# Association of Body Composition and Sarcopenia with NASH in Obese Patients

**DOI:** 10.3390/jcm10153445

**Published:** 2021-08-03

**Authors:** Sophia Marie-Therese Schmitz, Lena Schooren, Andreas Kroh, Alexander Koch, Christine Stier, Ulf Peter Neumann, Tom Florian Ulmer, Patrick Hamid Alizai

**Affiliations:** 1Department of General, Visceral and Transplantation Surgery, RWTH Aachen University Hospital, Pauwelsstr. 30, 52074 Aachen, Germany; lena.schooren@rwth-aachen.de (L.S.); akroh@ukaachen.de (A.K.); uneumann@ukaachen.de (U.P.N.); fulmer@ukaachen.de (T.F.U.); palizai@ukaachen.de (P.H.A.); 2Department of Gastroenterology, Digestive Diseases and Intensive Care Medicine, RWTH Aachen University Hospital, Pauwelsstr. 30, 52074 Aachen, Germany; akoch@ukaachen.de; 3Obesity Center NRW, Sana Kliniken, Krankenhausstr. 42, 50354 Hürth, Germany; Christine.Stier@Sana.de; 4Department of Surgery, Maastricht University Medical Center, P. Debyelaan 25, 6229 HX Maastricht, The Netherlands

**Keywords:** NAFLD, NASH, body composition, bioelectrical impedance analysis, BIA

## Abstract

Obese patients often suffer from sarcopenia or sarcopenic obesity (SO) that can trigger inflammatory diseases including non-alcoholic steatohepatitis (NASH). Sarcopenia and SO can be diagnosed through measuring parameters of body composition such as skeletal muscle mass (SMM), skeletal muscle index (SMI) and fat mass (FM) obtained by bioelectrical impedance analysis (BIA). The aim of this study was to assess the relationship of body composition and NASH in patients with obesity. A total of 138 patients with obesity that underwent bariatric surgery were included in this study. BIA was used to estimate body composition. A liver biopsy was taken intraoperatively and histological assessment of NASH was performed. A total of 23 patients (17%) were classified as NASH and 65 patients (47%) met the criteria for borderline NASH. Body mass index (BMI) was significantly higher in patients with NASH compared to borderline NASH and no NASH (56.3 kg/m^2^ vs. 51.6 kg/m^2^ vs. 48.6 kg/m^2^, *p* = 0.004). Concerning body composition, FM, but also SMM and SMI were significantly higher in patients with NASH (*p*-values 0.011, 0.005 and 0.006, resp.). Fat mass index (FMI) and weight-adjusted skeletal muscle index (SMI_weight) failed to reach statistical significance (*p*-values 0.067 and 0.661). In patients with obesity, higher FM were associated with NASH. Contrary to expectations, SMM and SMI were also higher in patients with NASH. Therefore, higher body fat, rather than sarcopenia and SO, might be decisive for development of NASH in patients with obesity.

## 1. Introduction

The decrease of skeletal muscle mass (sarcopenia) is a widely spread condition leading to multiple inflammatory diseases including non-alcoholic fatty liver disease (NAFLD) and its progressive form non-alcoholic steatohepatitis (NASH) [1,2,3,4,5,6]. The link between sarcopenia and NASH might be due to similar pathophysiological mechanisms [2,4,7,8,9]. NAFLD and NASH are increasingly attracting attention as their prevalence is rising and they account for a growing number of liver transplantations [10,11]. Higher amounts of body fat can be seen as predisposing factors for the development of NAFLD and NASH [12,13,14]. Therefore, a population especially vulnerable for NAFLD and NASH are patients with obesity, who naturally suffer from excess fat masses (FM) [15]. Contrary, the role of decreased skeletal muscle mass (SMM), (sarcopenia) for development of NASH is the subject of discussions. For diagnosis of sarcopenia, the relative SMM, as indicated by skeletal muscle index (SMI), is used most widely [4,8,15]. In lean patients, the SMI has been described to be inversely associated with NAFLD [6,16,17]. However, mechanisms of sarcopenia appear to differ in patients with and without obesity [18]. In patients with obesity, a condition named sarcopenic obesity (SO) has been described, that is characterized by high FM coupled with low SMM [18,19]. One underlying mechanism might be the release of pro-inflammatory cytokines in obese adipose tissue which leads to an inflammatory state [20] and further to SO, where SMM is not only diminished in quantity, but also in quality [21]. As the prevalence of sarcopenia is high in the general population, so is the prevalence of SO in an overweight and obese population, reaching a prevalence of 15–23% in patients with obesity [18,19]. SO has severe effects on patients’ health and triggers morbidity and mortality [1,22]. However, recent studies suggest a correlation of SMM with body mass index (BMI), and therefore higher SMM in patients with obesity [23]. The role of insulin resistance, which is triggered by decreased SMM, might therefore also be different in patients with obesity [24,25]. Measurement of body composition in patients with obesity provides intrinsic challenges. Body compartments behave differently in obesity and most measurements lack validation data [26]. Bioelectrical impedance analysis (BIA) provides an opportunity to assess body composition and sarcopenia by estimation of FM, fat-free mass (FFM) and SMM [27,28,29,30]. BIA-derived values have been cross-validated with dual-energy X-ray absorptiometry, computed tomography and magnetic resonance imaging [28,31]. Analysis with BIA has the advantage of being non-invasive, easy to conduct and is low in cost [21]. In patients with obesity, BIA-derived estimations of body composition appear to be better associated with body fat than simple calculation of body mass index (BMI) [32].

However, data on association of sarcopenia and SO with NASH in patients with obesity are sparse. The aim of this study was to investigate the relationship between body composition assessed by BIA and histologically proven NASH in patients with obesity.

## 2. Results

### 2.1. Clinical and Histological Characteristics

A total of 138 patients (71% female) participated in the study. Mean age was 44 years and mean BMI was 51 kg/m^2^. Prevalence of type 2 diabetes mellitus (T2DM) was 26% (*n* = 36) in this patient cohort. In total, 50 patients (36%) showed no signs of NASH (NAS-Score 0–2), while 23 patients (17%) were classified as definite NASH (NAS-Score 5–8) according to NAS score [33]. For a summary of the different components of the NAS-score, see Table 1. Sex did not show a statistically significant association with NASH in this study (*p* = 0.223). There was no statistically significant difference in age for patients with and without NASH (*p* = 0.124). For further characteristics of the study population see Table 2.

### 2.2. Body Composition and Liver Disease

One-way ANOVA showed a statistically significant difference in patients without NASH, borderline and definite NASH for BMI (means: no NASH: 48.6 kg/m^2^, borderline NASH: 51.6 kg/m^2^, definite NASH: 56.3 kg/m^2^, *p* = 0.004), body weight (means: no NASH: 138.0 kg, borderline NASH: 149.5 kg, definite NASH: 165.2 kg, *p* = 0.001), and the BIA-derived values fat free mass (FFM) (means: no NASH: 70.4 kg, borderline NASH: 75.9 kg, definite NASH: 82.5 kg, *p* = 0.018) and fat mass (FM) (means: no NASH: 67.9 kg, borderline NASH: 73.6 kg, definite NASH: 82.7 kg, *p* = 0.011) (see Figure 1, Figure 2 and Figure 3).

The estimated skeletal muscle mass (SMM) and skeletal muscle index (SMI) also showed statistically significant differences between groups with higher scores for NASH in comparison to no NASH (SMM means: no NASH: 31.0 kg, borderline NASH: 33.4 kg, definite NASH: 37.5 kg, *p* = 0.005; SMI means: no NASH: 10.8 kg/cm^2^, borderline NASH: 11.4 kg/cm^2^, definite NASH: 12.5 kg/cm^2^, *p* = 0.006). When adjusted for weight, SMI_weight did not show a significant difference between groups (means: no NASH: 0.08, borderline NASH: 0.08, definite NASH: 0.08, *p* = 0.662). Fat-free mass index (FFMI) was statistically significant between groups and showed an increase in case of definite NASH (means: no NASH. 24.4 kg/cm^2^, borderline NASH: 26 kg/cm^2^, definite NASH: 27.4 kg/cm^2^, *p* = 0.026), while fat mass index (FMI) did not differ between no NASH, borderline and definite NASH (means: no NASH: 23.8 kg/cm^2^, borderline NASH: 25.6 kg/cm^2^, definite NASH: 28 kg/cm^2^, *p* = 0.067). Phase angle (PhA) was not different in patients with NASH in comparison to the other groups (means: no NASH: 5.3°, borderline NASH: 5.3°, definite NASH: 5.3°, *p* = 0.959). For a summary of all tested values and scores see Table 2.

SMM and SMI showed a positive correlation with body weight (r = 0.701, *p* < 0.0001 and r = 0.664, *p* < 0.0001) (see Figure 4 and Figure 5).

SO as assessed by SMI_weight tertile did not show a significant association with NASH (20.9% NASH in the SO group vs. 15.2% NASH in the non-SO group; χ^2^, *p* = 0.88) (see Figure 5).

Insulin resistance was significantly different in patients with no, borderline and definite NASH (HOMA index means: no NASH: 6.9, borderline NASH: 9.8, definite NASH: 25.3, *p* = 0.000), while levels of glycated hemoglobin (HbA1c) did not differ between groups (means: no NASH: 5.8%, borderline NASH: 6.4%. definite NASH: 6.6%, *p* = 0.060).

HOMA Index showed a weak but significant correlation with SMI (r = 0.252, *p* = 0.016), but not with SMI_weight. (r = −0.09, *p* = 0.386) (Figure 6 and Figure 7).

## 3. Discussion

The relation of sarcopenia with NAFLD and NASH has been subject of recent discussions [1,5,7,9,16,17,34,35,36]. This connection is especially interesting in patients with obesity and sarcopenic obesity (SO), as high amounts of visceral fat have been described to be predisposing for NAFLD and NASH [13,15]. The aim of this study was to assess the association of body composition and NASH in patients with obesity.

As expected, we found an association between higher fat mass (FM), assessed by bioelectrical impedance analysis (BIA) and biopsy-proven NASH in this study. Interestingly, we could also demonstrate a significantly higher fat-free mass (FFM) and skeletal muscle mass (SMM) in patients with NASH. These findings may seem contradictory at first. However, in patients with obesity, these conditions coexist [23]. In our study, this might be explained by an association of a higher overall BMI with biopsy proven NASH in this study.

In line with our results, Ko et al. found visceral fat to be a strong predictor of NAFLD diagnosed by ultrasonography in a cross-sectional study [13].

Interestingly and against expectation, skeletal muscle index (SMI) was significantly higher in patients with NASH in this study. However, weight-adjusted SMI (SMI_weight) did not show a significant difference between patients with and without NASH. Low SMI has been described as the main marker for sarcopenia [17,37]. Several recent studies found an association between low SMI in patients with NAFLD in comparison to patients without NAFLD [4,7,8,16,36]. However, BMI in most studies was normal and diagnosis of NAFLD often relied on non-invasive methods. Of interest, definitions of SMI and sarcopenia appear to differ widely. Some researchers calculate SMI by dividing SMM by weight [17], while others divide by height [16]. This is especially important in patients with obesity, where weight shows a disproportionate relation to height. Even more, cut-off values for diagnosis of sarcopenia are lacking for patients with obesity. These findings highlight the importance to distinguish between height- and weight-adjusted SMI when comparing to other studies. Neither weight- nor height-adjusted SMI did however show the anticipated inverse association with NASH in this study, suggesting that sarcopenia in patients with obesity might not be the decisive factor for developing NASH. Confirming this, a large prospective cohort study found an increased risk for incident NAFLD in patients with overweight and obesity with higher FM at baseline, while lower SMI did not show a significant association with incident NAFLD during a mean follow-up of 48.5 months [15].

Another explanation for higher SMM and SMI in patients with NASH might be a metabolic phenotype with an unhealthy distribution of fat in patients with obesity. In metabolically unhealthy obese patients, skeletal muscle lipid content is higher than in metabolically healthy obese patients [38]. Not only muscle quantity, but also—quality is important for development of insulin resistance and inflammatory liver diseases [38,39]. The BIA-derived PhaseAngle (PhA) has been described as a measurement for muscle quality and nutritional status [21,40]. While SMM was different in patients with or without NASH in our study, PhA was not. This finding is supported by one study, where alterations of the PhA were predictive for NAFLD in patients with a BMI < 30 kg/m^2^, but not >30 kg/m^2^ [34]. These findings further support the idea, that mechanisms of sarcopenia might differ in patients with and without obesity, as generally higher PhA can be seen as protective from inflammatory and liver diseases [40,41].

Due to lacking cut-off values for diagnosis of sarcopenia in patients with obesity, patients with the lowest tertile of SMI in this study were classified as sarcopenic obese (SO). Congruent with the previous results, we did not find any significant differences for patients with or without SO and NASH.

Insulin resistance as measured by HOMA-IR was significantly higher in patients with NASH in this study in comparison to patients without NASH. Insulin resistance is an important parameter for the development of NAFLD and NASH [14]. Contrary to previously reported studies [24,25], HOMA-IR showed a weak but positive correlation with SMI. However, this correlation proved unsignificant when SMI was adjusted for weight. This furthermore underlines the central finding of this study that mechanisms of sarcopenia might differ in patients with and without obesity. There are several limitations to this study. First, there are sparse data on the validity of bioelectrical impedance analyses in patients with a BMI > 40 kg/m^2^ and specific cut-off values for the assessment of sarcopenia in patients with obesity are lacking [42]. However, BIA-derived estimation of body composition has already been used in patients with obesity and appears to correctly reflect changes in body composition after bariatric surgery [43,44,45]. Additionally, fat mass is rather under- and not over-estimated in individuals with obesity when assessed by BIA, which does not question the overall results of this study [42]. Furthermore, the results of our study are limited by the rather small number of patients included. Estimation of body composition with BIA additionally suffers from measure variability due to electrode placement. Notwithstanding these limitations, this is, to our knowledge, the largest study to compare BIA-derived data on body composition with histologically proven NAFLD and NASH in patients with obesity so far. BIA is non-invasive and easily conductible with results available immediately after measurement. It is, therefore, a valuable tool in daily, routine clinical practice. Further research with focus on repeated liver biopsies and BIA-measurements should validate the findings of this study.

## 4. Materials and Methods

Patients with obesity undergoing bariatric surgery were included into this prospective cohort study. Patients included met the indications for bariatric surgery: BMI > 40 kg/m^2^ or BMI > 35 kg/m^2^ with associated comorbidities. Patients under 18 years of age or with a history of alcohol consumption were excluded from this study. Written informed consent was obtained from each patient prior to enrolment. Ethical approval was obtained from our local Ethics Committee (RWTH Aachen University, EK 312/11).

During the week prior to operation, bioelectrical impedance analysis (BIA) was performed by a trained examiner using a Nutriguard MS multifrequency impedance analyzer (Data Input GmbH, Pöcking, Germany). In brief, two adhesive electrodes are applied to the skin of the right hand and foot while the patient is lying in a supine position. An electrical current of 50 kHz is produced by the BIA generator and the body resistance is measured by the attached electrodes. As intracellular and extracellular fluids behave differently, body composition can be estimated from the measured body resistance [29,30]. Body weight (BW) can be divided into fat mass (FM) and fat free mass (FFM). Both parameters can be adjusted for height squared and are subsequently labeled fat mass index (FMI) and fat free mass index (FFMI).

Estimation of skeletal muscle mass (SMM) from BIA values was performed as described by Janssen et al. using the formula: SMM = 0.401 height^2^/R50 + 3.825 gender − 0.071 age + 5.102 [31].

Skeletal muscle index (SMI) was calculated by dividing SMM (kg)/height^2^ (cm).

SMI was adjusted for weight (SMI_weight) by dividing SMM (kg) by body weight (kg) as described by Moon et al. [8]. Patients were classified into SMI_weight tertiles to compare patients with and without sarcopenic obesity as described previously [15]. The patients in the lowest SMI_weight tertile were classified as sarcopenic-obese (SO).

In addition, during the week prior to operation, blood was drawn after an overnight fast and blood parameters were measured. HOMA-IR was calculated by multiplying fasting insulin with fasting glucose, divided by 405.

During the bariatric operation, a liver wedge biopsy was performed from the left liver lobe. Liver specimens were evaluated by a single blinded hepato-pathologist and classified according to NAS classification by Kleiner et al. [33]. One-way ANOVA was used to compare mean values between groups. For multiple comparison, two-way ANOVA and Tukey multiple comparison test was used. A Chi-square test was used to compare dichotomous groups. A *p* < 0.05 was considered statistically significant. All statistical analysis was performed using IPM SPSS Statistics software v25 (IBM, Armonk, NY, USA) and GraphPad Prism v9 (GraphPad Software Inc., La Jolla, CA, USA).

## 5. Conclusions

This is to our knowledge the first study to compare body composition as assessed by BIA and histological diagnosis of NASH in patients with obesity. FM was significantly higher in patients with NASH in comparison to those without histologically proven NASH. Against expectations, parameters reflecting higher muscle mass (SMM and SMI) were also higher in patients with NASH. However, SMI_weight did not show a significant difference in patients with and without NASH, which questions the general applicability of parameters for assessment of sarcopenia and SO in patients with obesity. In conclusion, we hereby provide indications that total body fat and BMI might be more important for diagnosis of NASH in patients with obesity than sarcopenia and SO.

## Figures and Tables

**Figure 1 jcm-10-03445-f001:**
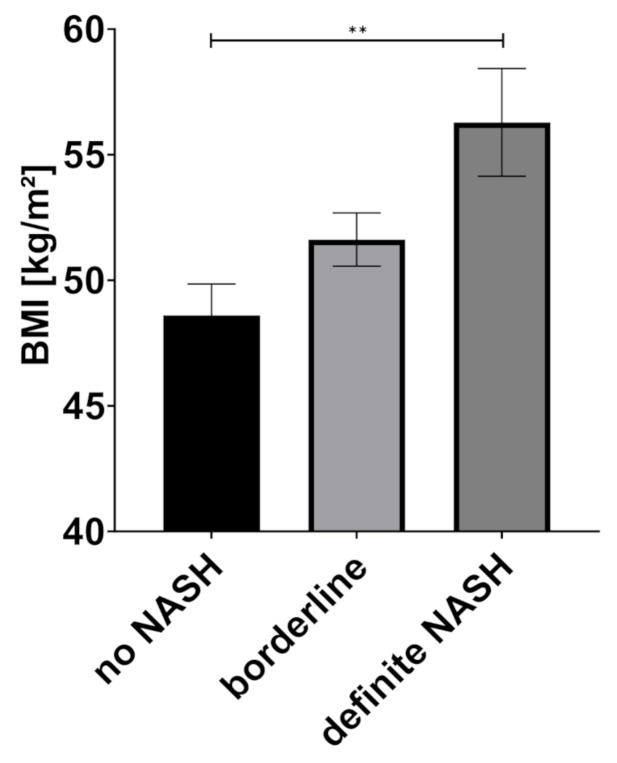
Body mass index (BMI) according to NAS classification, data indicated as mean (SEM), ** *p* = 0.0025, Abbreviations: BMI: body mass index, NASH non-alcoholic steatohepatitis.

**Figure 2 jcm-10-03445-f002:**
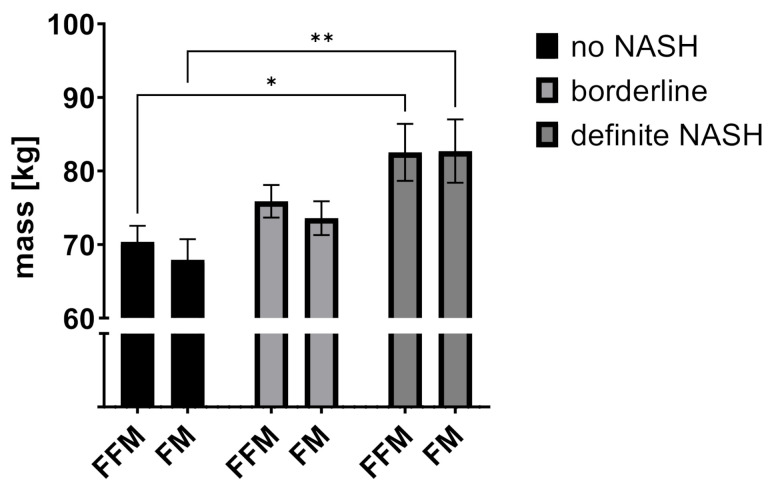
Fat mass and fat-free mass according to NAS subclasses. Data indicated as mean (SEM). Adjusted *p*-values * *p* = 0.0237, ** *p* = 0.0042 (Tukey’s multiple comparisons test). Abbreviations: FFM: fat-free mass; FM: fat mass; NASH: non-alcoholic steatohepatitis.

**Figure 3 jcm-10-03445-f003:**
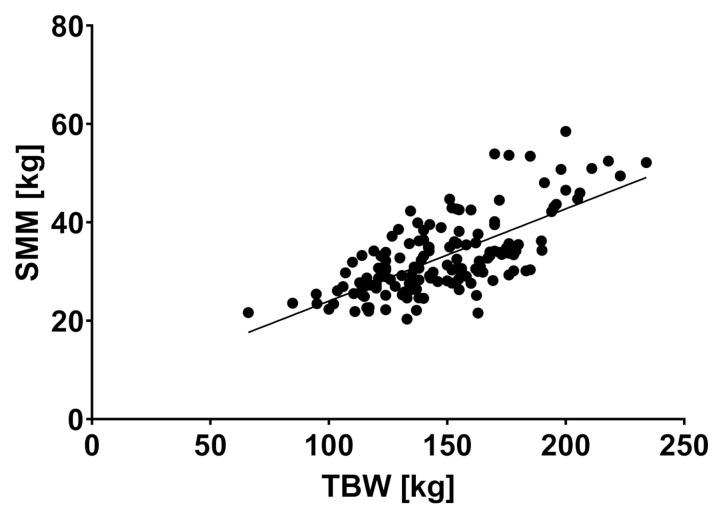
Skeletal muscle mass and total body weight showed a significant correlation, r = 0.701, *p* < 0.0001. Abbreviations: SMM: skeletal muscle mass, TBW: total body weight.

**Figure 4 jcm-10-03445-f004:**
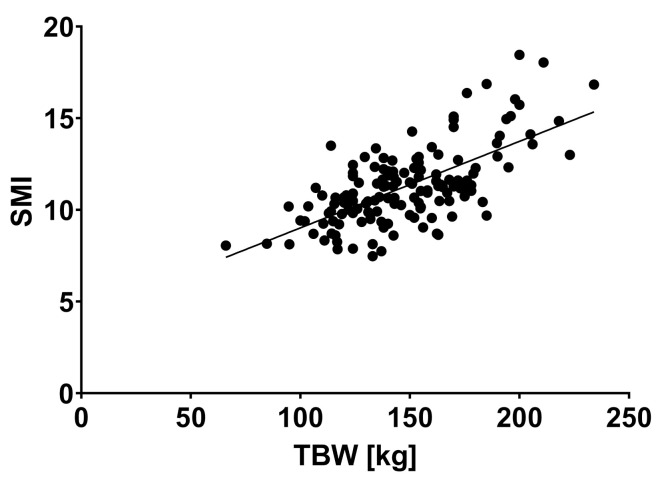
Skeletal muscle index and total body weight showed a significant correlation, r = 0.664, *p* < 0.0001). Abbreviations: SMI: skeletal muscle index, TBW: total body weight.

**Figure 5 jcm-10-03445-f005:**
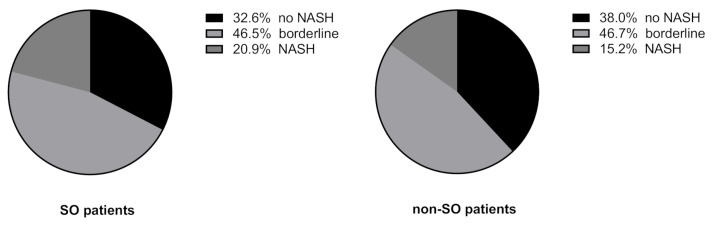
NAS classes according to presence of sarcopenic obesity (SO), defined by the lowest tertile in weight-adjusted SMI. (χ^2^, *p* = 0.8847). Abbreviations: NASH: non-alcoholic steatohepatitis, SO: sarcopenic obesity.

**Figure 6 jcm-10-03445-f006:**
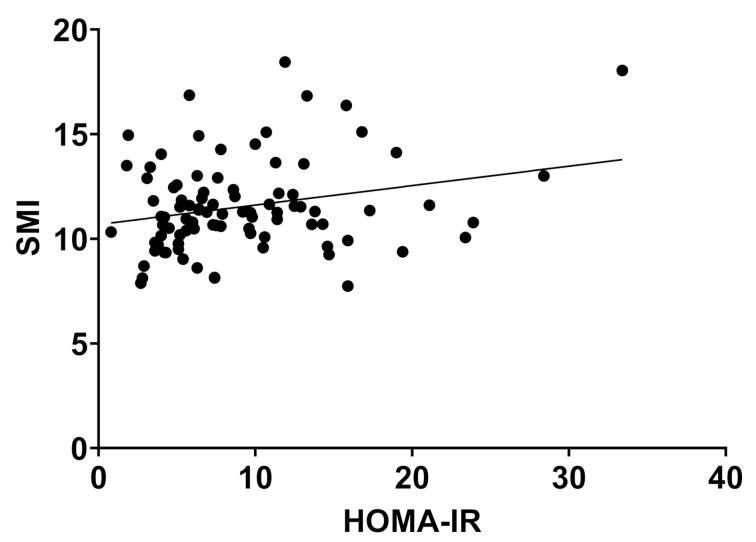
Skeletal muscle index and HOMA-IR showed a weak but significant correlation, r = 0.252, *p* = 0.016). Abbreviations: SMI: skeletal muscle index, HOMA-IR: homeostasis model assessment—insulin resistance.

**Figure 7 jcm-10-03445-f007:**
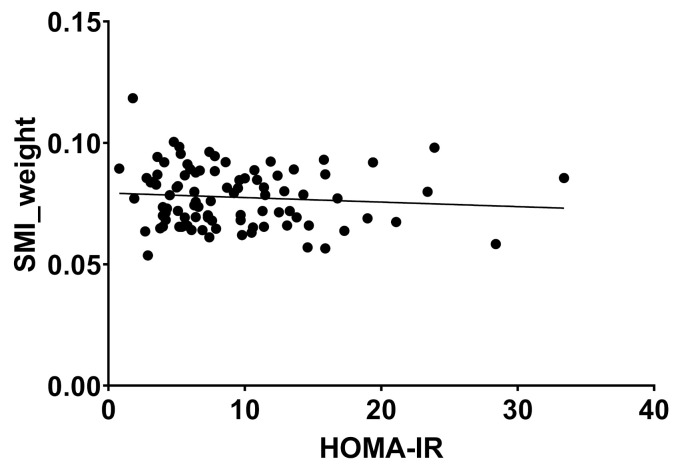
Weight-adjusted skeletal muscle index and HOMA-IR showed no significant correlation, r = −0.09, *p* = 0.386). Abbreviations: SMI: skeletal muscle index, HOMA: homeostasis model assessment—insulin resistance.

**Table 1 jcm-10-03445-t001:** Specifications of NAS characteristics; data indicated as *n* (%). Abbreviations: NAS: NAFLD activity score; NASH: non-alcoholic steatohepatitis.

		No NASH		Borderline NASH		Definite NASH	
		Count	*n* %	Count	*n* %	Count	*n* %
**Steatosis**	0: <5%	22	44.0	1	1.5	0	0.0
	1: 5–33%	26	52.0	28	43.1	2	8.7
	2: 34–4: 66%	2	4.0	29	44.6	8	34.8
	3: >66%	0	0.0	7	10.8	13	56.5
**Ballooning**	0: none	43	86.0	17	26.2	1	4.3
	1: few ballooned	7	14.0	44	67.7	16	69.6
	2: many ballooned	0	0.0	4	6.2	6	26.1
**Inflammation**	0: none	27	54.0	7	10.8	0	0.0
	1: <2 foci	23	46.0	49	75.4	7	30.4
	2: 2–4 foci	0	0.0	9	13.8	12	52.2
	3: >4 foci	0	0.0	0	0.0	4	17.4

**Table 2 jcm-10-03445-t002:** Characteristics of patients according to NAS classification; * indicates statistical significance applying ordinary one-way ANOVA. Abbreviations: ALT: alanine-aminotransferase; APRI: AST to platelet ratio index; AST: aspartate-aminotransferase; BMI: body mass index; WC: waist circumference; PhA: phase angle; TBW: total body water; FFM: fat-free mass; FFMI: fat-free mass index; FM: fat mass; FMI: fat-free mass index; kcal: kilocalories; HbA1c: glycated hemoglobin A1c; HOMA-IR: homeostasis model assessment; SMM: skeletal muscle mass; SMI: skeletal muscle index; SMI_weight: skeletal muscle index adjusted for weight.

	No NASH (*n* = 50)		Borderline NASH (*n* = 65)		Definite NASH (*n* = 23)		
	Mean	SD	Mean	SD	Mean	SD	*p*-Value
Age (years)	41.8	11.6	45.9	11.2	42.4	10-5	0.124
BMI (kg/m^2^)	48.6	8.8	51.6	8.5	56.3	10.3	0.004 *
Weight (kg)	138.0	29.3	149.5	27.9	165.2	31.2	0.001 *
WC (cm)	141.2	20.3	149.1	12.9	155.6	22.5	0.068
AST	22.7	6.0	28.9	17.2	33.4	11.1	0.003 *
ALT	24.5	7.9	38	30.5	44.3	19.8	0.001 *
APRI	0.08	0.03	0.11	0.1	0.12	0.05	0.030 *
Trigylcerides	134.9	75.0	175.1	86.9	187.6	83.6	0.016 *
Cholesterol	194.9	31.9	191.0	35.4	180.2	36.8	0.252
HbA1c (%)	5.8	1.3	6,4	1.6	6.6	1.7	0.060
HOMA-IR	6.9	4.6	9.8	5.7	21.0	25.3	0.000 *
PhA (°)	5.3	0.9	5.3	0.9	5.3	0.8	0.959
TBW (liter)	51.5	11.3	55.6	13.1	60.4	13.6	0.018 *
FFM (kg)	70.4	15.4	75.9	17.9	82.5	18.6	0.018 *
FM (kg)	67.9	19.8	73.6	18.5	82.7	20.6	0.011 *
FMI (kg/cm^2^)	23.8	7.3	25.6	6.8	28.0	7.9	0.067
FFMI (kg/cm^2^)	24.4	4.2	26.0	4.9	27.4	4.2	0.026 *
SMM (kg)	31.0	6.9	33.4	8.1	37.5	8.2	0.005 *
SMI (kg/cm^2^)	10.8	1.9	11.4	2.2	12.5	1.9	0.006 *
SMI_weight	0.08	0.0	0.08	0.0	0.08	0-0	0.661

## Data Availability

Data supporting reported results can be obtained on request from the authors.

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
