# Peer review of "Association of Body Composition and Sarcopenia with NASH in Obese Patients"

_jcm, 2021, doi:10.3390/jcm10153445_

Round 1
Reviewer 1 Report
This is a very interesting study of BIA in NASH patients. However, overall, statistical methods are inappropriate. The authors should seek advice from a statistician.
- Lines 19 through 24 of the Abstract are duplicates of the first sentence line 14-19. Please delete it.
- In this study, the authors did not analysis according to sex. I think Table 1 is meaningless and this information can be contained in Table 2.
- In Tables 1 and 2, the authors used the mean and standard error of means (SEM) to display the background of the subjects, but if the purpose is to show the distribution, the standard deviation should be used instead of the SEM.
- In Figure 2, there seems to be a problem of multiple comparisons. How did the authors adjust for this?
Author Response
This is a very interesting study of BIA in NASH patients. However, overall, statistical methods are inappropriate. The authors should seek advice from a statistician.
Thank you very much for your time and efforts in reviewing our paper. There are certainly more sophisticated methods for statistical analysis in this patient cohort. However, due to the clinical characteristics of this paper, we based our analysis mainly on comparison of means. The statistical methods used in this paper are basic but sound.
- Lines 19 through 24 of the Abstract are duplicates of the first sentence line 14-19. Please delete it.
Thank you very much, this has been resolved. We apologize for this inconvenience.
- In this study, the authors did not analysis according to sex. I think Table 1 is meaningless and this information can be contained in Table 2.
We deleted table 1. All information can be found in table 2.
- In Tables 1 and 2, the authors used the mean and standard error of means (SEM) to display the background of the subjects, but if the purpose is to show the distribution, the standard deviation should be used instead of the SEM.
We changed the table accordingly. All SEM were now replaced by SD.
- In Figure 2, there seems to be a problem of multiple comparisons. How did the authors adjust for this?
This has indeed not been stated clearly: we used 2-way ANOVA for comparison of FM and FFM and Tukey‘s multiple comparisons test for post hoc analysis in this case. This has been changed in the material and methods section as well as in the figure. We apologize for this.
Reviewer 2 Report
I red with interest the manuscript by Schmitz and colleagues aiming to investigate the association between body composition and NASH in obese patients.
Major revisions
- Clinical and histological information of the study population are missing. The authors grouped the study cohort according to the NAS score but they did not describe the single components of the NAS (hepatic steatosis, lobular inflammation and ballooning). Which is the median value of hepatic fat content? Similarly, clinical important information such as transaminases values, glucose and lipid profile data, physical activity information, are missing.
- It should be interesting to assess insulin resistance in this patients. Which is the prevalence of IFG/IGT/type 2 diabetes in this population? And what about the correlation between insulin resistance and sarcopenia?
- In this population of obese subjects without comorbidities, metabolically healty, sarcopenia is unlikely to develop because obese subjects tend to have more muscle mass to support their weight. BIA is imprecise in obese subjects, as stated in the limitation of the study and the association between sarcopenia and NASH is strange considering that in this population, NASH patients are those with a higher BMI. In fact, the SMI-weight parameter, the most appropriate parameter used for the evaluation of skeletal muscle index in obese subjects, did not differ according to the severity of NASH in this population.
- What about genetic predisposition? It could be interesting to assess the PNPLA3 polymorphism
- The conclusions are to strong considering the small number of cases and the methods used to evaluate sarcopenia in obese subjects.
Author Response
I red with interest the manuscript by Schmitz and colleagues aiming to investigate the association between body composition and NASH in obese patients.
Dear Reviewer,
Thank you very much for your time to review our paper. We appreciate your comments and tried to resolve everything accordingly.
Major revisions
- Clinical and histological information of the study population are missing. The authors grouped the study cohort according to the NAS score but they did not describe the single components of the NAS (hepatic steatosis, lobular inflammation and ballooning). Which is the median value of hepatic fat content? Similarly, clinical important information such as transaminases values, glucose and lipid profile data, physical activity information, are missing.
Thank you very much again for this remark. We added a table with the single components of the NAS. Furthermore, we added clinical information, transaminase values, lipid and glucose profiles to the patient cohort characteristics.
- It should be interesting to assess insulin resistance in this patients. Which is the prevalence of IFG/IGT/type 2 diabetes in this population? And what about the correlation between insulin resistance and sarcopenia?
Prevalence of type 2 diabetes mellitus was 26% in our study cohort. We added this information to the results section. We added further information on insulin resistance and sarcopenia and performed a correlation analysis between HOMA-IR and SMI. We added this new and interesting information to the discussion section as well.
- In this population of obese subjects without comorbidities, metabolically healty, sarcopenia is unlikely to develop because obese subjects tend to have more muscle mass to support their weight. BIA is imprecise in obese subjects, as stated in the limitation of the study and the association between sarcopenia and NASH is strange considering that in this population, NASH patients are those with a higher BMI. In fact, the SMI-weight parameter, the most appropriate parameter used for the evaluation of skeletal muscle index in obese subjects, did not differ according to the severity of NASH in this population.
This is indeed an important remark. Furthermore and interestingly, HOMA-IR was correlated positively with SMI but not with SMI_weight. This is another proof, that mechanisms of sarcopenia are different in patients with obesity and that therefore this patient group cannot be evaluated applying the same parameters as patients without obesity.
- What about genetic predisposition? It could be interesting to assess the PNPLA3 polymorphism
This is indeed a very interesting research question. Sadly our patients did not provide consent for genetic analysis and therefore we cannot provide these information.
- The conclusions are to strong considering the small number of cases and the methods used to evaluate sarcopenia in obese subjects.
We tried to resolve this by stating repeatedly that our findings provide some evidence but need to be confirmed in larger studies.
Round 2
Reviewer 1 Report
The authors has made the appropriate corrections. I have no further comments.
Reviewer 2 Report
I would like to thank you the authors for providing new data.
I have no other comments